

# Association of metabolically healthy obesity and elevated risk of coronary artery calcification: a systematic review and meta-analysis

Yu-wen Hsueh[1,*], Tzu-Lin Yeh[2,3], Chien-Yu Lin[4,8], Szu-Ying Tsai[5], Shu-Jung Liu[6], Chi-Min Lin[2] and Hsin-Hao Chen[2,7,8,*]

[1] Department of Internal Medicine, MacKay Memorial Hospital, Taipei, Taiwan
[2] Department of Family Medicine, Hsinchu MacKay Memorial Hospital, Hsinchu, Taiwan
[3] Institute of Epidemiology and Preventive Medicine, National Taiwan University, Taipei, Taiwan
[4] Department of Pediatrics, Hsinchu MacKay Memorial Hospital, Hsinchu, Taiwan
[5] Department of Family Medicine, MacKay Memorial Hospital, Taipei, Taiwan
[6] Department of Medical Library, MacKay Memorial Hospital, Tamsui Branch, New Taipei City, Taiwan
[7] MacKay Junior College of Medicine, Nursing, and Management, Taipei, Taiwan
[8] Department of Medicine, MacKay Medical College, New Taipei City, Taiwan
[*] These authors contributed equally to this work.

Corresponding author
Hsin-Hao Chen, 2033@mmh.org.tw

## ABSTRACT

**Background**. Metabolically healthy obesity (MHO) is defined as obesity with less than two parameters of metabolic abnormalities. Some studies report that MHO individuals show similar risk of cardiovascular disease (CVD) compared with metabolically healthy non-obese (MHNO) individuals, but the results are conflicting. Coronary artery calcium (CAC) reflects the extent of coronary atherosclerosis and is a useful tool to predict future risk of CVD. The objective of this meta-analysis was to investigate whether MHO is associated with elevated risk of CAC.

**Method**. We searched Cochrane, PubMed, and Embase up to April 19, 2019. Prospective cohort and cross-sectional studies examining the association between MHO subjects and CAC were included with MHNO as the reference. Pooled odds ratio (OR) and 95% confidence interval (CI) were calculated using random-effect models. Subgroup analysis and meta-regression were applied to define possible sources of heterogeneity. We conducted this research following a pre-established protocol registered on PROSPERO (CRD 42019135006).

**Results**. A total of nine studies were included in this review and six studies with 23,543 participants were eligible for the meta-analysis. Compared with MHNO subjects, MHO had a higher odds of CAC (OR 1.36, 95% CI [1.11 to 1.66]; $I^2 = 39\%$). In the subgroup analysis, the risk associated with MHO participants was significant in cohort studies (OR = 1.47, 95% CI [1.15,1.87], $I^2 = 0\%$), and borderline significant in cross-sectional studies. The risk of CAC was also significant in MHO participants defined by Adult Treatment Panel III (ATP III) (OR = 1.55, 95% CI [1.25,1.93], $I^2 = 0\%$). The univariate meta-regression model showed that age and smoking status were possible effect modifiers for MHO and CAC risk.

**Conclusion**. Our meta-analysis showed that MHO phenotypes were associated with elevated risk of CAC compared with MHNO, which reflects the extent of coronary

atherosclerosis. People with obesity should strive to achieve normal weight even when only one metabolic abnormality is present.

## INTRODUCTION

Worldwide obesity rates have nearly tripled in the last 40 years, and to date more than one third of the global population is overweight or obese (*Collaboration, 2016*). Obesity is associated with a higher risk of incident metabolic syndrome, which in turn is associated with a two-fold increase in the risk of CVD and a 1.5-fold increase in the risk of all-cause mortality (*Bastien et al., 2014*; *Mongraw-Chaffin et al., 2016*; *Mottillo et al., 2010*). Therefore, it has become a major global health burden (*Seidell & Halberstadt, 2015*).

However, not all obese individuals have metabolic abnormalities. This group includes those without insulin resistance, dyslipidemia, glucose intolerance, hypertension, or high inflammatory status (*Velho et al., 2010*). Recent research has focused on a phenotype of obese individuals, termed the MHO, referring to obese subjects with less than 2 risk parameters of the metabolic syndrome (except accounting for waist circumference) and elevated homeostatic model for assessing insulin resistance (HOMA-IR) and elevated c-reactive protein (CRP) levels, which is now being widely used (*Stefan, Schick & Haring, 2017*). Based on different criteria of metabolic syndrome as well as sex and age, MHO accounts for as much as 30%–40% of the obese adult population (*Stefan et al., 2013*; *Velho et al., 2010*; *Wildman et al., 2008*; *Zheng, Zhou & Zhu, 2016*). Earlier epidemiological studies indicate that MHO individuals are not at increased risk of CVD after a short term follow-up (*Appleton et al., 2013*; *Hamer & Stamatakis, 2012*; *Hosseinpanah et al., 2011*). However, the link between MHO and CVD remains controversial. Some meta-analyses have reported that obese individuals with metabolic phenotypes considered 'healthy' are still at increased risk of CVD after sufficient long-term follow up (*Kramer, Zinman & Retnakaran, 2013*; *Mirzababaei et al., 2019*). On the other hand, no uniform definition of MHO was established during these studies. Most studies defined MHO as a combination of four common metabolic criteria: blood pressure, high-density lipoprotein cholesterol (HDL-C), triglycerides (TG), and fasting plasma glucose. Other components, such as HOMA-IR or CRP, were not readily adopted (*Roberson et al., 2014*). One meta-analysis in 2015 also indicated that stricter criteria may be needed to identify benign obesity phenotypes (*Eckel et al., 2016*).

Coronary artery calcium (CAC), detected by coronary multidetector computed tomography (MDCT), reflects the extent of subclinical atherosclerosis and suggests the presence of CVD (*Budoff et al., 2006*). Previous studies that examined the link between MHO and CAC found inconsistent results and several limitations existed with regards to the association of MHO and CAC progression (*Chang et al., 2014*; *Echouffo-Tcheugui et al., 2019*; *Kang et al., 2017*; *Rhee et al., 2013*; *Yoon et al., 2017*). Some of them were cross-sectional studies, conducted in Asian populations, which did not allow us to examine the
temporal relationship between MHO and CAC progression (*Chang et al., 2014*; *Jung et al., 2014*; *Rhee et al., 2013*; *Sung et al., 2014*). Absence of a common definition for MHO made it difficult to clarify whether the MHO phenotype is harmful to coronary arteries. Since CAC is an important indicator of atherosclerotic disease, defining the extent of calcification in coronary arteries, the objective of this study was to systematically and comprehensively explore the relationship between MHO and CAC risk in comparison to MHNO participants.

## METHODS

This systematic review and meta-analysis were conducted following a pre-established protocol registered on PROSPERO (CRD 42019135006) and reported in accordance with the Preferred Reporting Items for Systematic Reviews and Meta-analyses (PRISMA) guidelines (*Shamseer et al., 2015*) (Table S1). All authors declare that there is no conflict of interest regarding the publication of this study. The research did not receive any specific grants from funding agencies in the public, commercial or not-for-profit sectors.

### Data sources and searches

We performed a systematic literature search using the Medline, EMBASE, and Cochrane library databases supplemented with the manual review of the reference list of obtained articles up to April 19, 2019. We used different combinations of the following Medical Subject Headings (MeSH) terms: ("coronary calcification" or "coronary calcium" or "coronary atherosclerosis" or "vascular calcification") and ("Metabolically Benign Obesity", "Metabolically Healthy Obesity", "obesity phenotype"). Full search strategies are shown in Table S2. Articles were selected if the title or abstract indicated that the study analyzed the association between CAC and MHO. Two authors, Hsin-Hao Chen and Yu-Wen Hsueh, independently conducted the searches and any disagreements were resolved via discussion with a third author, Chien-Yu Lin. We also performed a manual search of references from relevant publications, as well as previous reviews and meta-analyses. Parameter ranges for language, year of publication, article type, and participant characteristics including age were not limited to enable a relatively comprehensive search.

### Study selection

We included all eligible publications that satified our inclusion criteria: (1) published observational cohort and cross-sectional studies investigating MHO and CAC conducted on adults; (2) classification of obesity and non-obesity by waist circumference (WC) in addition to the body mass index (BMI) shown in the pre-established protocol; (3) reporting of criteria used to define metabolically healthy/unhealthy (MH/MUH) phenotypes; (4) presence of a reference group with metabolically healthy non-obese or normal weight individuals; (5) reporting of the presence, extent, new development, or progression of CAC, assessed by electron beam computed tomography (EBCT) or MDCT. Studies were excluded if they (1) were duplicate publications, (2) topically irrelevant, (3) did not compare MHO and CAC with MHNO or metabolically healthy normal weight (MHNW) people, or (4) were literature reviews, republished data, case reports, dissertations, or conference abstracts.

## Data extraction and quality assessment

We extracted the following information: first author's name, year of publication, description of the study population (country, number of participants/cases per MHO phenotype, mean age, sex proportion, duration of follow-up in cohort studies), study design, definition of metabolically healthy and obese, diagnostic methods of CAC, adjusted variables, and main outcome presented with OR (95% CI) using MHNO as the reference.

The quality of the included studies was assessed independently by two authors, Chien-Yu Lin and Szu-Ying Tsai, using the Newcastle Ottawa Scale (NOS). The quality assessment tool rates each study in three domains—selection, comparability, and outcome—using a star system ranging from zero to nine stars in cohort studies and from zero to ten stars in cross-sectional studies (*Zeng et al., 2015*). If two authors disagree, the decision was made by the third author, Chi-Min Lin. A study was considered to be of high quality if the cohort study obtained at least six stars and the cross-sectional study obtained at least seven stars.

## Statistical analysis and data synthesis

We calculated pooled odds ratios (ORs) with 95% CIs for estimating the risk of CAC progression in MHO compared with MHNO. We used statistical computing software R, Version 1.1.456, primarily using the Comprehensive R Archive Network (CRAN) package "metagen" for our meta-analysis (*R Core Team, 2013*). We employed a random effects model using DerSimonian and Laird's method under an assumption of non-identical true effect sizes (*DerSimonian & Laird, 1986*). The results were presented as forest plots. Heterogeneity among studies was quantified by Cochran's $Q$ test and $I^2$ statistics (*Higgins & Thompson, 2002*) and further explored by pre-specified subgroup analyses. We also used meta-regression models to test contributions of effect modifiers, consisting of age, sex, and smoking status (*Rothman, 2012*). With regards to the statistical significance interpretation, we do not treat $p$-values dichotomously according to the previous studies. We consider the $p$-values as graded measures of the strength of evidence (*Amrhein, Greenland & McShane, 2019*; *Amrhein, Korner-Nievergelt & Roth, 2017*). We performed a sensitivity analysis by excluding one study at a time to measure its impact on the robustness of the results. Publication bias was evaluated by funnel plots and Egger's tests with a significant publication bias defined by a $p$ value $< 0.1$ (*Egger et al., 1997*).

## RESULTS

### Description of studies and quality assessment

The flowchart of article selection is shown in Fig. 1. A total of 9 observational studies including cohort (*Kang et al., 2017*; *Kowall et al., 2019*; *Yoon et al., 2017*) and cross-sectional (*Chang et al., 2014*; *Echouffo-Tcheugui et al., 2019*; *Jung et al., 2014*; *Khan et al., 2011*; *Rhee et al., 2013*; *Sung et al., 2014*) studies were included in our systematic review. One article that combined MHO and metabolically healthy overweight (MHOW) people into one group was excluded from our meta-analysis because it did not comply with our criteria (*Khan et al., 2011*). Another two articles were excluded because they were series from the same institutions with possible duplicate participants (*Chang et al., 2014*; *Rhee et al., 2013*). Finally, six studies fulfilled all the inclusion criteria for the meta-analysis

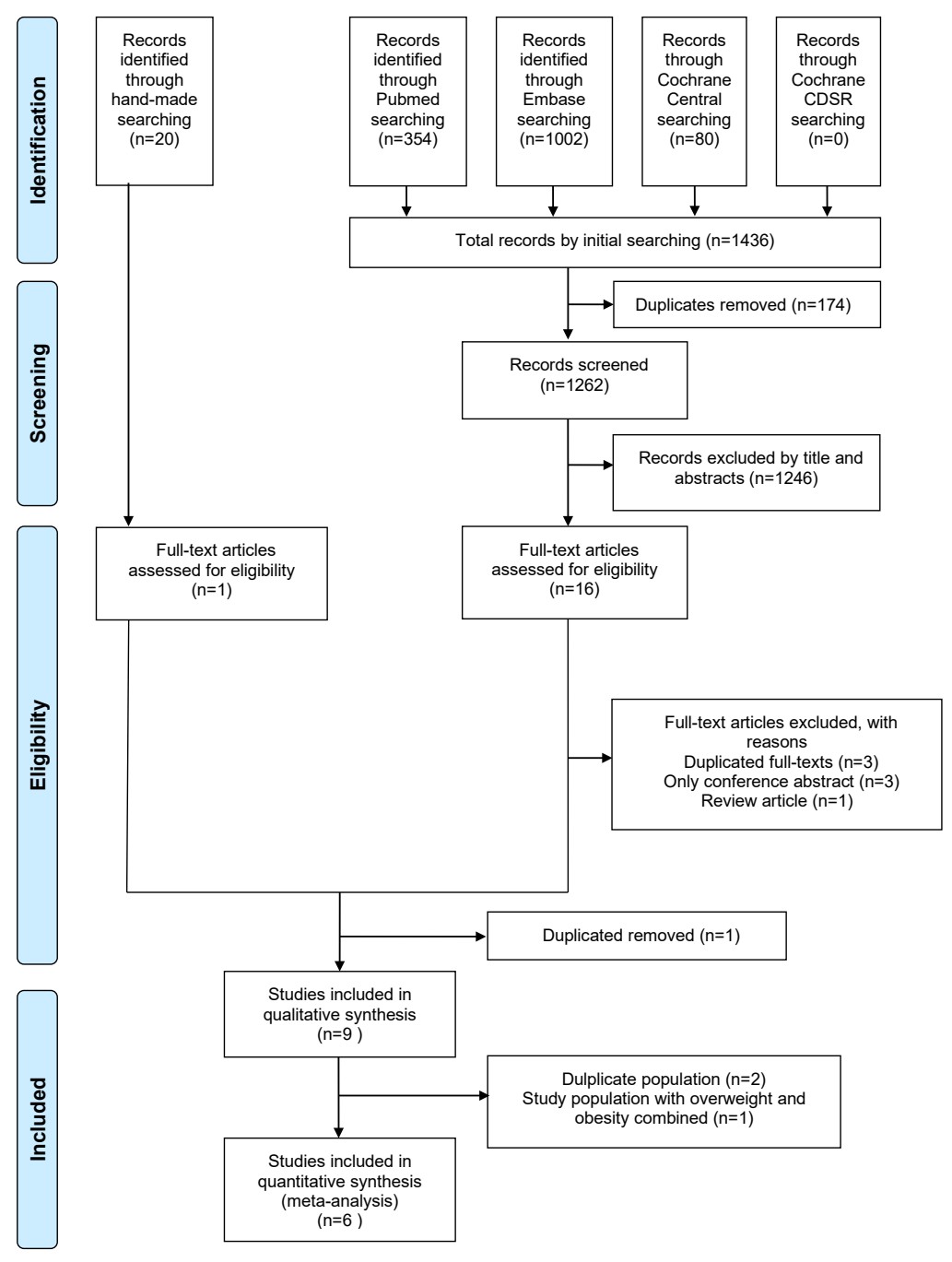

**Figure 1** Flowchart of the study selection process.

(*Echouffo-Tcheugui et al., 2019*; *Jung et al., 2014*; *Kang et al., 2017*; *Kowall et al., 2019*; *Sung et al., 2014*; *Yoon et al., 2017*).

The general demographic characteristics of subjects in the nine included studies in the systematic review are summarized in Table 1. Among the 62,909 participants, 16.8% were

**Table 1  Characteristics of included studies.**

| Study | Participants | Obesity definition: BMI (kg/m$^2$) or WC (cm) | Definition of metabolic healthy (criteria and numbers of components | Adjusted variables | Diagnostic criteria and main results presented by MHO compared with MHNO with OR (95%CI) | NOS |
|---|---|---|---|---|---|---|
| Khan et al. (2011) | USA, Study of Women's Health Across the Nation, N = 475, 100% women, 55% in MHO, 50.9 y/o, cross sectional | BMI ≥25 (include overweight and obese) | NCEP-ATP III (except WC) + CRP ≥ 3.0 mg/dL MHO: <3 | Age, site of recruitment, education, race, smoking | CACS ≥10 assessed by CT MHO/MHOW vs MHNW: OR 2.38(1.2–4.7) | 9 |
| Rhee et al. (2013) | Korea, part of Kangbuk Samsung Health Study N = 24063, 82% women, 18.2% in MHO, 41.3 y/o, cross sectional | BMI ≥25 or WC (≥90 in men, ≥80 in women) | Wildman criteria MHO: ≤2 | Age, sex, smoking, SBP, FBS, TC, TG, hs-CRP, Ca | CACS assessed by CT: OR for different CACS categories CAC 1-10: 1.23(1.03-1.48) 11-100: 1.16(0.97-1.38) 101-400: 1.43(1.00- 2.06) >400: 1.79(0.84–3.79) | 7 |
| Sung et al. (2014) | Korea, Kangbuk Samsung Hospital N = 14384, 17.5% women, 6.6% in MHO, 41.3 y/o, cross sectional | BMI ≥25 or WC (≥90 in men, ≥80 in women) | JIS criteria MHO: =0 | Age, sex, smoking, alcohol, exercise, LDL-C, HOMA-IR, CRP, TSH | CACS assessed by CT (CACS >0 ) MHO: OR 0.93(0.67–1.31) | 8 |
| Chang et al. (2014) | Korea, part of Kangbuk Samsung Health Study, N = 14828, 25.8% women, 21.9% in MHO, 39.3 y/o, cross sectional | BMI ≥25 | NCEP-ATP III (except WC) and HOMA-IR MHO: =0 and HOMA-IR<2.5 | Age, sex, smoking, alcohol, exercise, education | CACS assessed by CT MHO vs MHNW: a. CACS ratio: 2.26 (1.48-3.43) b. prevalence ratio of 1.CACS>80: 1.67 (1.09–2.56) 2.CACS1-80: 1.39 (1.15–1.67) | 7 |
| Jung et al. (2014) | Korea, Asan Medical Center, N = 4009, 17.7% women, 14.7% in MHO, 52.3 y/o, cross sectional | BMI ≥25 | Wildman criteria MHO: ≤1 | Age, sex, WC, alcohol, smoking, exercise, diabetes, SBP, FBS, ALT, GGT, uric acid, LDL-C, HDL-C, hs-CRP, HOMA-IR. | CACS assessed by CT CACS >0: 1.38 (1.04-1.82) 0.1-100: 1.32 (0.98-1.79) >100: 1.69 (1.03- 2.78) | 8 |
| Yoon et al. (2017) | Korea, Seoul National University Hospital, N = 1218, 27.6% women, 15.6% in MHO I, 54.6 y/o, 3.75 yrs f/u | BMI ≥25 | NCEP-ATP III (except WC) MHO I: ≤1 MHO II: =0 | Age, sex | CACS progression (any Agatston score increase >0 at f/u) by CT MHO I: OR: 1.65 (1.14-2.40) MHO II: OR: 1.20 (0.51–2.78) | 8 |

**Table 1** (*continued*)

| Study | Participants | Obesity definition: BMI (kg/m²) or WC (cm) | Definition of metabolic healthy (criteria and numbers of components | Adjusted variables | Diagnostic criteria and main results presented by MHO compared with MHNO with OR (95%CI) | NOS |
|---|---|---|---|---|---|---|
| *Kang et al. (2017)* | Korea, Asan Medical Center, $N = 1240$, 18.3% women, 22.7% in MHO, 54.2 y/o, 2.9 yrs f/u | BMI ≥25 | NCEP-ATP III (except WC) MHO: ≤1 | Age, sex, WC, alcohol, smoking, exercise, baseline CAC score, LDL-C, hs-CRP, and f/u interval | CACS progression (any ≥2.5 units between the baseline and final square root of the CACS) by CT MHO: OR 1.45(0.93–2.25) | 9 |
| *Kowall et al. (2019)* | Germany, Heinz Nixdorf Recall Study, $N = 1585$, 60.5% women, 10.0% in MHO, 58.3 y/o, 5.1 yrs f/u | BMI ≥30 | NCEP-ATP III (except WC) MHO I: ≤1 MHO II: =0 | Age, sex, smoking, physical activity, education | Categorical variable for annual absolute CAC change (CAC at f/u minus CAC at baseline): ≥100 AU; 10-99 AU; <10 AU OR calculated from ordinal logistic regression models (vs MHNW) MHO I: 1.24 (0.78-1.97) MHO II: 1.29 (0.43–3.9) | 9 |
| *Echouffo-Tcheugui et al. (2019)* | United States, Framingham Offspring Study, $N = 1107$, N/A % women, N/A % in MHO, 45 y/o, cross sectional | BMI ≥30 | NCEP-ATP III (except WC) MHO: ≤1 | Age, sex, smoking | CAC score >100, assessed by CT MHO: OR 1.94(1.18–3.19) | 7 |

**Notes.**

ALT, alanine aminotransferase; AU, Agatston units; BMI, body mass index; BP, blood pressure; CAC, coronary artery calcification; CACS, coronary artery calcification score; Ca, calcium; CI, confidence interval; cm, centimeter; CT, computed tomography; CRP, C reactive protein; FBS, fasting blood sugar; f/u, follow/up; GGT, gamma glutamyl transpeptidase; HDL-C, high-density lipoprotein Cholesterol; HOMA-IR, homeostatic model of the assessment of insulin resistance; hs-CRP, high-sensitivity C-reactive protein; JIS, Joint Interim Statement; LDL-C, Low Density Lipoprotein Cholesterol; MHO, metabolically health obesity; MHNO, metabolically healthy non-obese; MHNW, metabolically healthy normal-weight; N/A, not available; NCEP-ATP III, National Cholesterol Education Program-Adult Treatment Panel III; NOS, Newcastle-Ottawa Scale; OR, odd ratio; SBP, systolic blood pressure; TC, total cholesterol; TG, triglyceride; TSH, Thyroid-stimulating hormone; WC, waist circumference; yrs, years.

MHO, with a mean age of 42.2 years and a 46.6% female proportion. All included studies were published after 2011 and most were conducted in Korea (*Chang et al., 2014*; *Jung et al., 2014*; *Kang et al., 2017*; *Rhee et al., 2013*; *Sung et al., 2014*; *Yoon et al., 2017*), followed by the USA (*Echouffo-Tcheugui et al., 2019*; *Khan et al., 2011*) and Germany (*Kowall et al., 2019*). One study was restricted to women (*Khan et al., 2011*). Sample size varied substantially among studies, ranging from 475 to 24,063 participants. The duration of follow-up ranged from 2.9 to 5.1 years in the three cohort studies (*Kang et al., 2017*; *Kowall et al., 2019*; *Yoon et al., 2017*), while the remaining were cross-sectional studies. (*Chang et al., 2014*; *Echouffo-Tcheugui et al., 2019*; *Jung et al., 2014*; *Khan et al., 2011*; *Rhee et al., 2013*; *Sung et al., 2014*).

Obesity was defined by a BMI ≥ 25 in the Korean studies (*Chang et al., 2014*; *Jung et al., 2014*; *Kang et al., 2017*; *Rhee et al., 2013*; *Sung et al., 2014*; *Yoon et al., 2017*) and ≥30 in the German and USA studies (*Echouffo-Tcheugui et al., 2019*; *Khan et al., 2011*;
*Kowall et al., 2019*). Metabolic status was defined by ATP III criteria only in four studies (*Echouffo-Tcheugui et al., 2019*; *Kang et al., 2017*; *Kowall et al., 2019*; *Yoon et al., 2017*) with additional assessment of HOMA-IR or CRP in two studies (*Chang et al., 2014*; *Khan et al., 2011*). The other three studies used Joint Interim Statement (JIS)(*Sung et al., 2014*) and Wildman criteria (*Jung et al., 2014*; *Rhee et al., 2013*).

For evaluating outcomes, the three cohort studies used CAC progression as expressed by a categorical change or change in the absolute coronary artery calcification score (CACS) (*Kang et al., 2017*; *Kowall et al., 2019*; *Yoon et al., 2017*). The other cross-sectional studies used OR at different CACS cut-off points or categories with the MHNO group as the reference (*Chang et al., 2014*; *Echouffo-Tcheugui et al., 2019*; *Jung et al., 2014*; *Khan et al., 2011*; *Rhee et al., 2013*; *Sung et al., 2014*).

All three cohort studies achieved at least eight out of nine stars on the NOS quality assessment scale (shown in Table S1). The cross-sectional studies scored from seven to nine out of ten stars, indicating that all included studies were of good quality. The detailed scores are shown in Tables S3-1 and S3-2.

## Results of the meta-analysis

For CAC risk analysis, six observational studies with 23,543 subjects were pooled for the meta-analysis (*Echouffo-Tcheugui et al., 2019*; *Jung et al., 2014*; *Kang et al., 2017*; *Kowall et al., 2019*; *Sung et al., 2014*; *Yoon et al., 2017*). Participants with MHO had significantly higher odds of CAC than those with MHNO (OR = 1.36, 95% CI [1.11, 1.66], $I^2 = 39\%$, Fig. 2). Due to the underlying heterogeneity in MHO definitions and article types, we analyzed data in subgroups. Compared with MHNO participants, those with MHO had significantly higher odds of CAC progression in the cohort studies (OR = 1.47, 95% CI [1.15, 1.87], $I^2 = 0\%$, forest plot shown in Fig. 2), and it is very likely that MHO is associated with CAC prevalence in the cross-sectional studies(OR = 1.31, 95% CI [0.90, 1.91], $I^2 = 69\%$, forest plot shown in Fig. 2). Compared with MHNO participants, those with MHO had higher odds of CAC according to metabolic status as defined by the National Cholesterol Education Program-Adult Treatment Panel III (NCEP- ATP III) (OR = 1.55, 95% CI [1.25, 1.93], $I^2 = 0\%$, forest plot shown in Fig. 3), but not according to other definitions (OR = 1.15, 95% CI [0.78, 1.69], $I^2 = 68\%$, forest plot shown in Fig. 3).

We also performed meta-regression analyses for potential effect modifiers, consisting of sex, age, and smoking status. Univariate meta-regression showed no statistically significant effect modification with proportion of women ($p = 0.31$) and was borderline significant with age and smoking status ($p = 0.08$, $p = 0.058$; bubble plots shown in Figs. S1–S3). When we performed a sensitivity analysis by excluding one article in which the reference group was MHNW (*Kowall et al., 2019*) rather than non-obese, the result still remained robust (OR = 1.39, 95% CI [1.10, 1.76], $I^2 = 50\%$). We also excluded one study at a time and re-calculated the overall effect estimate, and the results were not changed. No significant publication bias was detected by Egger's tests ($p = 0.47$) and there was no substantial asymmetry in the funnel plot (Fig. S4).

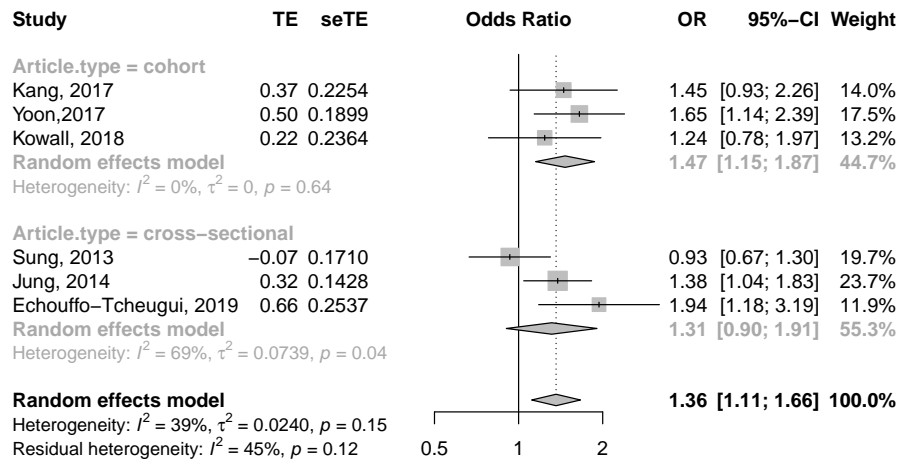

**Figure 2** Forest plot of CAC risk, comparing participants with metabolically healthy obesity as those with metabolically healthy non-obesity, with subgroup analysis by article design. CAC, coronary artery calcification; CI, confidence interval; OR, odds ratio; SE, standard error; TE, treatment effect.

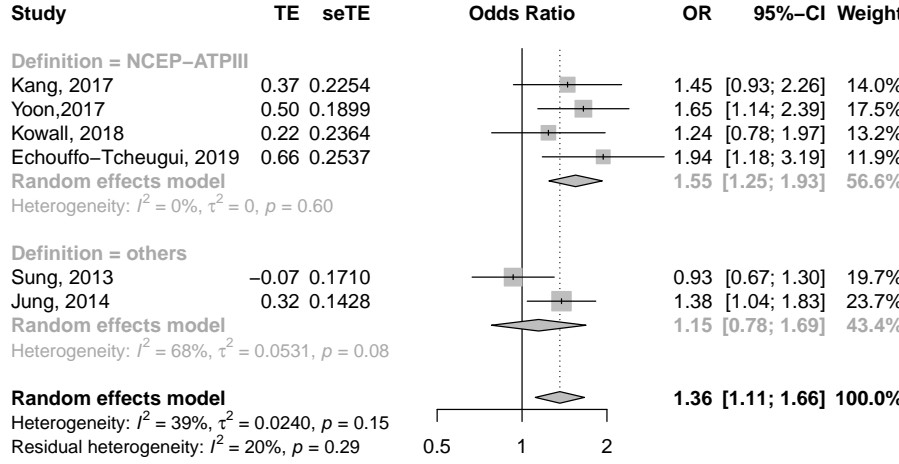

**Figure 3** Forest plot of CAC risk, comparing participants with metabolically healthy obesity as those with metabolically healthy non-obesity, with subgroup analysis by definition of metabolic health. ATP III, Adult Treatment program III; CAC, coronary artery calcification; CI, confidence interval; JIS, Joint Interim Statement; OR, odds ratio; SE, standard error; TE, treatment effect.

## DISCUSSION

The meta-analysis showed that compared to individuals with MHNO, those with MHO phenotypes had a significantly higher odds of CAC, especially in cohort studies and when defined by NCEP-ATP III criteria. Age and smoking status were possible effect modifiers for MHO and CAC risk.

Our study is the first systematic review and meta-analysis to show that MHO phenotype increases the risk of CAC progression, which reflects the extent of coronary atherosclerosis

and quite possibly indicates increased future risk of CVD. Some previous studies have reported an association between MHO and CVD risk. One meta-analysis of fourteen prospective studies in 2013 showed that MHO people had increased risk of CVD with a pooled relative risk (RR) of 2.00 (95% CI [1.79–2.24]) compared with healthy normal-weight individuals. The results appeared much stronger during the long-term follow-up period of more than 15 years (*Fan et al., 2013*). Another review in 2013 showed that MHO people are at elevated risk of CVD (RR = 1.24, 95% CI [1.02–1.55]) when only studies with 10 or more years of follow-up were considered. (*Kramer, Zinman & Retnakaran, 2013*) Another meta-analysis in 2016 showed that MHO phenotypes defined by different definitions of metabolic syndrome, insulin resistance, or other approaches, were at increased risk for CVD (*Eckel et al., 2016*). The significantly higher odds of CAC progression shown in the cohort studies included in our analysis agreed with these previous reviews that MHO phenotype may not be a benign condition.

In the subgroup analysis of our study, MHO participants in cohort studies showed a higher odds of CAC progression, and it is likely that MHO is associated with CAC prevalence in the cross-sectional studies. Of the studies included in our meta-analysis, three were cross-sectional and only offered a snapshot of subclinical coronary atherosclerosis in MHO (*Echouffo-Tcheugui et al., 2019*; *Jung et al., 2014*; *Sung et al., 2014*). Although two studies still showed higher odds of CAC (*Echouffo-Tcheugui et al., 2019*; *Jung et al., 2014*) compared with MHNO subjects, this material did not allow us to examine CAC progression over time. An original pilot study, which invited 88 consecutive community-based participants with a 3.5-year follow-up, indicated a regular CACS increase of 24% each year since baseline (*Maher et al., 1999*). Therefore, given a longer follow-up duration, we were able to detect the significant difference in CAC between MHO and MHNO.

Another source of heterogeneity is the definition of metabolic health, regarding which there is still a lack of consensus among MHO-related studies. Most studies included in our meta-analysis (*Echouffo-Tcheugui et al., 2019*; *Kang et al., 2017*; *Kowall et al., 2019*; *Yoon et al., 2017*) defined metabolic status following NCEP-ATP III, referring to the presence of any three of the following five criteria: (a) serum TG $\geq$150 mg/dl or under drug treatment; (b) serum HDL-C <40 mg/dl in men or <50 mg/dl in women or under drug treatment; (c) blood pressure of $\geq$130/85 mmHg or under drug treatment; (d) fasting plasma glucose $\geq$100 mg/dL or under drug treatment; and (e) abdominal obesity defined as a WC of $\geq$102 cm in men or $\geq$88 cm in women (*Expert Panel on Detection E & Treatment of High Blood Cholesterol in A, 2001*). All included studies using NCEP-ATP III excluded WC due to collinearity with BMI. The other criteria used to define MHO are the JIS and Wildman criteria. The latest JIS definition is similar to NCEP-ATP III, but with ethnicity-specific WC cut-points (*Alberti et al., 2009*). In the Wildman criteria (*Wildman et al., 2008*), metabolically healthy is defined as having less than two of the following six risk factors: (a) systolic blood pressure (SBP) $\geq$130 mmHg and/or diastolic blood pressure (DBP) $\geq$85 mmHg, or on antihypertensive treatment; (b) TG $\geq$150 mg/dl or use of lipid-lowering drugs; (c) fasting glucose $\geq$100 mg/dl or being treated for diabetes; (d) HDL-C < 40 mg/dL in men or <50 mg/dL in women; (e) HOMA-IR >90th percentile (>5.13 mole $\times$ $\mu$U/L$^2$); and (f) high-sensitivity C-reactive protein (hs-CRP) >90th percentile (>0.1 mg/L).

Wildman criteria also included HOMA-IR, the core concept of metabolic syndrome, and CRP, the best biomarker of vascular inflammation and predictor of CVD events (*Jeppesen et al., 2008*). These criteria modifications for metabolically healthy individuals also explained the heterogeneity in our meta-analysis. Despite the fact that heterogeneity in the subgroup of cohort studies reached zero ($I^2 = 0\%$), the definiton of CAC progression differed in the three included cohort studies. CAC progression was defined by absolute CAC change in two cohort studies (*Kang et al., 2017*; *Yoon et al., 2017*), and a categorical variable for annual CAC change was used in another study (*Kowall et al., 2019*). So more studies with more uniform standards for CAC evaluation were needed to verify our results.

Another issue is that we classified participants as metabolically healthy using at most one component (except WC) from the criteria in our study. However, a previous meta-analysis showed that compared with MHNW, MHO participants not expressing any of these metabolic factors showed no significantly increased CV risk (*Eckel et al., 2016*). In one large study by *Lassale et al. (2017)*, an increased risk of CHD was found in MHO individuals, when MH was defined by the metabolic syndrome criteria, but not when MH was defined by the absence of any cardiometabolic risk factor. The conclusion was consistent with the result of our three included studies (*Kowall et al., 2019*; *Sung et al., 2014*; *Yoon et al., 2017*), which showed those without any metabolic risk factor had no significantly higher risk of CAC. This may be one reason for conflicting results in previous studies. Moreover, studies show that CAC risk in MHO is higher than that in MHNO, but lower than that in metabolically unhealthy obesity (MUO) (*Kowall et al., 2019*; *Sung et al., 2014*). This message might be that improving one's metabolic profile may be worthwhile even if the person remains obese. MHO may be an intermediate step for people with MUO to reduce CAC risk. More studies are thus needed to better define metabolically healthy and benign obese phenotypes.

Age, and smoking status were possible effect modifiers in our study. Previous studies indicated that CAC risk increased with age, whereby the extent of CAC was greater in men compared with women up to the age of 60 (*Otsuka et al., 2014*). In contrast, smoking was independently related to lipid-rich plaques, contributing to an increased risk of plaque composition (*Kumagai et al., 2015*). The CARDIA (*Loria et al., 2007*) and the Heinz Nixdorf Recall (*Lehmann et al., 2014*) study show that current smoking was positively associated with CAC risk. This is consistent with our results from the univariate meta-regression, which indicated that impact of MHO on CAC risk decreased as the proportion of smokers increased. Future studies will help confirm that age, sex, and smoking status are potentially very important modifiers of the association effect between MHO and CAC risk.

Previous studies have reported the possible underlying mechanisms of MHO. In most obese subjects even without metabolic dysfunction, the adipocyte storage capacity may be exceeded (*Bluher, 2010*) and extra lipid may accumulate ectopically in visceral fat depots, liver, muscle, and $\beta$-cells. These locations are associated with the risk of developing CVD (*Roca-Rivada et al., 2015*). Inflammation in adipose tissue has been proposed as another key factor, as adipose tissue secretes bioactive peptides, adipokines, interleukin-6, and tumor necrosis factor-alpha, which affect multiple functions, such as immunity, insulin
sensitivity, angiogenesis, blood pressure, lipid metabolism, and hemostasis, all of which are linked with CAC (*Ronti, Lupattelli & Mannarino, 2006*). Therefore, obese subjects even without metabolic dysfunction should not be considered as benign phenotypes, but rather as having pre-metabolic syndrome, which is also correlated with a high risk of developing metabolic dysfunction-related cardiovascular comorbidities (*Rasaei et al., 2018*).

Lifestyle intervention is considered safe and effective at decreasing CV risk in obese individuals (*Stefan, Haring & Schulze, 2018*). In one review of MHO, weight loss of 5–10% may be enough for better cardiometabolic health, but more weight loss might be needed in higher BMI individuals (*Kantartzis et al., 2011*). In addition to weight loss, Mediterranean diet could be combined with weight loss programs for metabolic health (*Estruch et al., 2013*). And the absence of fatty liver is also a strong predictor for the regression from metabolically unhealthy to the metabolically healthy condition, which may be taken into account in the future studies.

Our study is the first review and meta-analysis to indicate the elevated risk of CAC in MHO compared with MHNO subjects. However, some limitations should be taken into account. First, only six studies were included in the meta-analysis and nine studies in our review, of which six were conducted in Korea (*Chang et al., 2014*; *Jung et al., 2014*; *Kang et al., 2017*; *Rhee et al., 2013*; *Sung et al., 2014*; *Yoon et al., 2017*) while the other three were from the USA and Germany (*Echouffo-Tcheugui et al., 2019*; *Khan et al., 2011*; *Kowall et al., 2019*). Another issue is that the sample size varied substantially, mainly among cross-sectional studies. Although they all scored highly on the NOS scale, more studies are needed from other countries to confirm our results. Second, the included studies differed in their definitions of metabolically healthy and obese. Although we performed subgroup analyses in which MHO defined by NCEP ATP-III with $I^2 = 0\%$ showed elevated risk of CAC compared with MHNO, more studies are needed in which metabolically healthy is defined by JIS or using a universally accepted definition. In other words, most of our included studies use BMI alone to define obesity (*Chang et al., 2014*; *Echouffo-Tcheugui et al., 2019*; *Jung et al., 2014*; *Kang et al., 2017*; *Khan et al., 2011*; *Kowall et al., 2019*; *Yoon et al., 2017*), which may not be sufficient to distinguish between fat and lean tissue. Adiposity measures between Asian and Caucasian populations cannot be easily compared, and BMI cut-offs separating normal weight, overweight and obesity differ between these populations, this limitation may have affected the results. A recent study suggested that using BMI alone is not appropriate (*Chrysant & Chrysant, 2019*). Thus further studies using or combing other indices are warranted. Third, three of the studies included in our meta-analysis were cross-sectional, precluding establishment of a temporal relationship. We performed a subgroup analysis by study design and synthesis of cohort studies that showed significantly higher odds of CAC progression in MHO. Fourth, with regards to statistical limitations, the recommended minimum number of studies suggested in order to conduct a meta-regression is ten (*Higgins & Green, 2011*). Although age and smoking were possible important effect modifiers of the association between MHO and CAC risk, there are only five studies included in the meta-regression. Therefore, we need more research to confirm the results.

CAC is an early indicator of atherosclerotic disease and a more accurate predictor of CVD than traditional risk factors. Another commonly used subclinical measure of CVD is common carotid intima medial thickness (CCA-IMT), an indicator of arterial wall thickness and atherosclerotic disease progression. Bobbioni-Harsch et al. performed a three-year progression evaluation showing significantly elevated CCA-IMT in MHO compared to normal body weight (*Bobbioni-Harsch et al., 2012*) which is consistent with our results.

## CONCLUSIONS

Our study showed a significant association between MHO and elevated risk of CAC, which in turn reflects the extent of coronary atherosclerosis. People with obesity should strive to achieve normal weight even when only one metabolic abnormality is present.

## ACKNOWLEDGEMENTS

Thanks to the MacKay Memorial Hospital librarian, Pei-jin Li, for examining the references. We would like to thank Anthony Abram for editing and proofreading this manuscript. We thank to the Wei-hsin Liang, physician of MacKay Memorial Hospital, for reviewing this manuscript.

### Funding
The authors received no funding for this work.

### Competing Interests
The authors declare there are no competing interests.

### Author Contributions
- Yu-wen Hsueh performed the experiments, prepared figures and/or tables, and approved the final draft.
- Tzu-Lin Yeh conceived and designed the experiments, analyzed the data, prepared figures and/or tables, and approved the final draft.
- Chien-Yu Lin and Shu-Jung Liu performed the experiments, authored or reviewed drafts of the paper, and approved the final draft.
- Szu-Ying Tsai and Chi-Min Lin analyzed the data, authored or reviewed drafts of the paper, and approved the final draft.
- Hsin-Hao Chen conceived and designed the experiments, performed the experiments, analyzed the data, prepared figures and/or tables, and approved the final draft.

### Data Availability
The collated data included in our meta-analysis is available in the Supplemental File.

## Supplemental Information

Supplemental information for this article can be found online at http://dx.doi.org/10.7717/peerj.8815#supplemental-information.

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
