# Peer review of "Association of metabolically healthy obesity and elevated risk of coronary artery calcification: a systematic review and meta-analysis"

_PeerJ, doi:10.7717/peerj.8815_

## Round 0.1 · original submission · Major Revisions

Dear authors,

Your paper has been assessed by several experts and all of them have indicated scientific merit in your work. However, there are several issues which you should address in a revised version of the text.

Best regards,
Dr Palazón-Bru

Reviewer 1 ·

Basic reporting

no comment

Experimental design

no comment

Validity of the findings

no comment

Additional comments

Hsueh et al. present results of a meta-analysis on metabolically healthy obesity and CAC. They confirm that MHO is associated with higher prevalence / risk of CAC than MHNO. Some points should be addressed by the authors:

1. The authors only compare MHO to MHNO / MHNW. However, also MUO (metabolically unhealthy obesity) is of interest. Studies show that cardiovascular risk in MHO is higher than the CVD risk in MHNO, but lower than the CVD risk of MUO. Thus, the message might be that improving one’s metabolic profile may be worthwhile even if the person remains obese. Striving to achieve normal weight – as the authors put it – and keeping normal weight is hard, so MUO may at least be a first intermediate goal which helps to reduce CVD risk (or risk of CAC progression).

2. In Kowall et al. (2019), there is not only a longitudinal analysis of the MHO CAC association, but also a cross-sectional one (table 2). Why didn’t the authors take this cross-sectional analysis into account?

3. The authors quote Stang (2010) as a reference for the Newcastle-Ottawa Scale. However, this reference is a harsh criticism of the NOS, and the author advises against using the NOS.

4. On page 15, the authors write that „age, sex, and smoking were important effect modifiers“. On page 14, they write that „univariate meta-regression showed no statistically significant effect modification with age and proportion of women (p=0.08, p=0.31)“. Does this mean that the authors do not support significance testing (what would deserve recognition)? If so, they should state this in the methods section. However, from their result on cross-sectional studies (OR=1.34 (95% CI: 0.91 – 1.96)), the authors conclude that a higher risk of CAC progression „was not detected in cross-sectional studies“ (also: p. 13, line 196). This is wrong because from the confidence interval it is very likely that MHO is associated with CAC prevalence. For problems of significance testing, please cf. Amrhein V et al. The earth is flat (p > 0:05): significance thresholds and the crisis of unreplicable research. PeerJ 2017; Amrhein V et al. Retire statistical significance. Nature 2019.

5. In the discussion, the authors mention that there is heterogeneity in the definiton of MHO. I suggest discussing that definitons of CAC progress also differ in the three included cohort studies.

6. P. 20, line 308: Kowall et al. also used waist circumference to define obesity.

7. P. 13: higher odds of CAC (not: higher risks of CAC), when ORs were estimated

8. Abstract, line 22: „MHO is defined as obesity without metabolic abnormalities“. This is not correct because some definitions allow the prevalence of 1 (or even 2) metabolic abnormalities.

·

Basic reporting

Please see the general comments for the author

Experimental design

Please see the general comments for the author

Validity of the findings

Please see the general comments for the author

Additional comments

In this meta-analysis the authors investigated whether MHO is associated with elevated risk of CAC. For this they included 3 cohort studies and 6 cross-sectional studies in their analysis. They found that MHO associated with elevated risk of CAC compared with MHNO.

Comments:
1. Lines 58-60: The statement ‘Recent research has focused on a phenotype of obese individuals, termed the MHO, which refers to obese subjects who fulfill the definition of obesity without meeting the criteria of metabolic syndrome.’ implicates that the suggested definition of metabolic health and metabolic syndrome are the same. However, that is not the case. In the first 2 papers that were published back-to-back in 2008 proposing that a distinct metabolically healthy phenotype exists, insulin sensitivity measured by a frequently sampled OGTT was the initial definition (Arch Intern Med. 2008 Aug 11;168(15):1609-16.) and in lack of such data Wildman et al. suggested defining metabolic health when having less than 2 risk parameters of the metabolic syndrome (except accounting for waist circumference) and elevated HOMA-IR and elevated CRP levels (Arch Intern Med. 2008 Aug 11;168(15):1617-24.). This definition is now being widely used (Cell Metab. 2017 Aug 1;26(2):292-300.). Most importantly, the definitions of metabolic syndrome and metabolic health for sure are different in the number of risk parameters that are allowed to be present when considering having a metabolically beneficial phenotype.
2. A limitation of this study is that most studies included in this meta-analysis were conducted in the Korean and, thus, in an Asian population. Because adiposity measures between Asian and Caucasian populations cannot be easily compared and BMI cut-offs separating normal weight, overweight and obesity differ between these populations, this limitation may have affected the results.
3. Another limitation that needs also to be addressed in the discussion is the fact that sample size varied substantially among studies.
4. Discussion, line 229: the statement ‘Another review in 2013, which included only studies with 10 or more years follow-up, also showed that MHO people are at elevated risk of all-cause mortality and CVD.’ is not correct, because that review also included studies with less than 10 years of follow-up and in the analysis including all studies, MHO was not associated with increased risk of all-cause mortality.
5. Furthermore, the authors should discuss the large study by Lassale et al. (Eur Heart J. 2018 Feb 1;39(5):397-406.) where an increased risk of CHD was found in MHO individuals, when MH was defined by the metabolic syndrome criteria, but not when MH was defined by the absence of any cardiometabolic risk factor.
6. Discussion, line 293: when the authors discuss clinical implications of MH in obesity, they should also address that this concept can be used to investigate how much weight loss and what other treatment options should be recommended in obesity, and that fatty liver is a strong predictor for the regression from the metabolically unhealthy to the metabolically healthy condition (Lancet Diabetes Endocrinol. 2018 Mar;6(3):249-258.).

Reviewer 3 ·

Basic reporting

In this manuscript, the authors performed a systematic review and meta-analysis of the presence or progression of coronary artery calcification in metabolically healthy obesity compared to metabolically healthy non-obesity or normal weight. This article gives a good introduction with relevant literature cited. The article structure and tables/figures are appropriate, and the raw data is shared. This article generally uses clear and professional language. I have a few suggestions to clarify language below:
1. Line 56: “section” should be changed to “group”
2. Line 78 “…several limitations concerning the association…” is a bit vague
3. Line 80 “Asia” should be changed to “Asian”
4. Line 160: Please clarify that the “included studies” you mention are the nine studies included in the systematic review, rather than the six studies included in the meta-analysis.
5. Line 267: “we classified participants as metabolically healthy using at least one component from the criteria in our study.” Should this say “at most one”? Also, from Table 1 it looks like the number of components needed for MHO classification varies from study to study. Could you clarify this?
6. Line 282: “impact of MHO on CAC risk decreased as the proportion of smokers increased.” Is this smokers in the MHO group, or in total across the MHO and control groups?
7. In the paragraph beginning on line 285, the first sentence suggests that the paragraph will talk about mechanisms of metabolically healthy obesity, but the paragraph goes on to talk about the mechanisms of disease in obesity. Perhaps consider revising.
8. Line 298: Authors state “only nine studies were included in our review,” which is slightly misleading since only six of those made it in to the meta-analysis.
9. Line 303-304: Duplicate sentence
10. Line 313: Please change “causal” to “temporal”, since neither cross-sectional nor cohort studies alone can show a causal relationship.

Experimental design

The design of the systematic review and meta-analysis is generally good. My comments are below:
1. The meta-regression analyses do not have a large enough sample size. The recommended minimum number of studies suggested in order to conduct a meta-regression is 10, and there are only 5 included in the meta-regression. This limitation should at least be mentioned, or the meta-regression should be omitted. Looking at figures S1 and S3, the slope of the line appears to be heavily influenced by a single influential study. The multivariable meta-regression should not be included in this study, as the R-squared value of 100% suggests extreme overfitting of the model to the data. There are 3 effects to estimate in the model and only 5 observations.
2. It is not explained why one of the 6 studies from the meta-analysis was not included in the meta-regression. Looking at the raw data, it appears that there are no values for smoking, age, or sex in the MHO group in the Echouffo-Tcheugui et al. 2019 study. However, these values are presented in Table 2 of Echouffo-Tcheugui et al. 2019, and so this study should be included in the meta-regressions and the raw data should be updated.
3. Smaller notes from within the text:
a. Line 196-7: the risk of “CAC progression” was said to be higher in cohort studies, but not in cross-sectional studies. It isn’t possible to measure CAC progression in cross-sectional studies, so this should be re-worded.
b. Line 259: Should “DBP<85” instead be “DBP>=85?”
c. Line 265-266: “These criteria modifications for metabolically healthy individuals explained the heterogeneity in our meta-analysis.” Please elaborate/provide evidence for how the criteria modifications explain the heterogeneity.
4. The protocol for systematic review was published online prior to analysis. However, the protocol for the systematic review reported in PROSPERO does not always match what is reported in the Methods section for this paper. These deviations from the protocol should be addressed in the manuscript. Some discrepancies include:
a. Line 111-112: Inclusion criteria for obesity definition includes obesity defined by Waist Circumference, whereas the registered protocol only allows BMI-defined obesity
b. Line 114-115: Includes studies which report “the presence, extent, new development, or progression of CAC”, whereas the protocol only mentions the progression of CAC. However, this may be due to the inclusion of cross-sectional studies (allowed in the original protocol), which by design cannot assess CAC progression.
c. Line 131-132: The cutoff for high quality studies in the PROSPERO protocol was at least 7 for all studies, whereas in the manuscript it is listed as at least 6 for cohort studies and at least 7 for cross-sectional studies.
d. The protocol says subgroup analysis will be run by country, but this was not done in the manuscript.
e. The protocol says sensitivity analysis will be run by excluding one study at a time and re-calculating the overall effect estimate, but this was not done in the manuscript.

Validity of the findings

The study is relevant and well-designed, but I have serious concerns about the validity of the findings as the manuscript is written currently.
1. There are many contradictions between the effect estimates from the individual studies reported in the raw data, in Table 1, and Figures 2-3. For example, Echouffo-Tcheugui et al. 2019 is reported as having an OR for the relevant effect measure of 1.94 (1.18-3.19) in Table 1, 1.94 (1.28-2.95) in Figures 2 and 3, and 1.94 (1.38-3.19) in the raw data. Another example: in Kowall, 2018, the effect estimate is reported as 1.24 (0.78-1.79) in Table 1, and 1.24 (0.78-1.97) in Figures 2-3 and the raw data. There are several other such inconsistencies, mostly with the upper bounds of the confidence intervals of the individual effect estimates. Since the confidence intervals are presumably used to calculate the standard errors for the effect estimates, it is likely that the overall meta-analyzed effect estimate reported in this study is not valid.

---

## Round 0.2 · Major Revisions

Still pending some modifications to improve your work. See the reviewers' comments for more information.

·

Basic reporting

No comment

Experimental design

No comment

Validity of the findings

No comment

Additional comments

Based on the fact, and also as mentioned in the response of the authors to the referee's comment number 5, that the risk of CAC was not elevated in subjects with MHO, when defined as having no metabolic risk parameter, compared to subjects with MHNO, the authors need to re-word the statement in the conclusion: 'People with obesity should strive to achieve normal weight even in the absence of metabolic abnormalities.' e.g. to 'People with obesity should strive to achieve normal weight even when only one metabolic abnormality is present.'

Reviewer 3 ·

Basic reporting

Please see "General comments for the author" section.

Experimental design

Please see "General comments for the author" section.

Validity of the findings

Please see "General comments for the author" section.

Additional comments

I appreciate the authors addressing the comments in my original review and believe they have adequately addressed nearly all suggestions I offered. I now understand the reason for some of the upper confidence limits being slightly different in Figures 2 and 3 from the original reported results in Table 1, and I agree with the authors that this minimal rounding error as a result of the meta-analysis is not cause for concern to the validity of the findings. In fact, I think the sentence added by the authors: “Another statistical issue is that the overall meta-analyzed effect estimate may not be so valid because the original confidence intervals are used to calculate the standard errors for the effect estimates. But we do not think the main result will be changed due to the statistical conversion” can be omitted, since the rounding error does not seem to affect the main result. Or, perhaps, it could be changed to explain that some confidence bounds are slightly different in Figures 2/3 compared to Table 1, and explain that it is due to rounding error because the ORs reported in the original studies are not so precise.

However, there is still one study which appears to be incorrect in the main analysis, and not due to rounding error. For Echouffo-Tcheugui et al. 2019, Table 1 appears to correctly report the OR and its confidence interval as 1.94 (1.18-3.19). However, in the raw data, it is reported as 1.94 (1.38-3.19), which appears to be a typographical error. This means that the standard error calculated for this study appears to have been calculated based on an incorrect and narrower confidence interval than the CI from the original study. In Figures 2 and 3, the OR and 95% CI are reported for Echouffo-Tcheugui et al. 2019 as being 1.94 (1.28-2.95), which is not similar to the correct CI (1.18-3.19), likely due to the 95% CI being calculated by the meta-analysis program on a too-small standard error from the incorrect confidence interval in the raw data (1.38-3.19).

If I am misunderstanding what happened here, please let me know, but this appears to be an error that may affect the final conclusions of the study. The overall meta-analyzed estimate, as well as the subgroup analyses, Figures 2 and 3, and the meta-regression and funnel plots, would also be affected and need to be re-done.


Finally, I have a few small wording/English language use suggestions:
Line 113: I am not sure if “besides” means “in addition to” here? I think the latter phrasing would be clearer if so.
Line 146: change both instances of “p-value” to “p-values”
Line 271: change “reached to zero” to “reached zero”
Line 274: change “standard” to “standards”
Line 309-310: change “obesity subjects” to “obese individuals” or similar
Line 322: change “which mainly existed” to “mainly”
Line 338: change “the statistical limitation” to “statistical limitations”
Line 340: change “smoke” to “smoking”
Line 342: change “researches” to “research”

---

## Round 0.3 · Minor Revisions

Still pending a minor modification suggested by one of the reviewers.

Reviewer 1 ·

Basic reporting

no comment

Experimental design

no comment

Validity of the findings

no comment

Additional comments

The authors have adequately responded to all my comments. I suggest they should drop a word on comment 1 in the discussion (MHO as an intermediate step for persons with MUO).

·

Basic reporting

No further comments

Experimental design

No further comments

Validity of the findings

No further comments

Additional comments

No further comments

Reviewer 3 ·

Basic reporting

No comment.

Experimental design

No comment.

Validity of the findings

On line 204, lower bound "1.23" should be "1.25" to match result in Figure 3.

Additional comments

Thank you for incorporating my suggestions, and best of luck with your publication!

---

## Round 0.4 · accepted · Accept

All the reviewers' concerns have been correctly addressed.

Reviewer 3 ·

Basic reporting

No comment

Experimental design

No comment

Validity of the findings

No comment

Additional comments

Thank you for incorporating my suggestions, and best of luck with the submission.